# Leisure Programmes in Hospitalised People: A Systematic Review

**DOI:** 10.3390/ijerph20043268

**Published:** 2023-02-13

**Authors:** Paula Adam-Castelló, Eva María Sosa-Palanca, Luis Celda-Belinchón, Pedro García-Martínez, María Isabel Mármol-López, Carlos Saus-Ortega

**Affiliations:** 1Department of Health La Ribera, Integrated Health Center of Sueca, 46410 Sueca, Spain; 2Nursing School La Fe, Adscript Centre of University of Valencia, 46026 Valencia, Spain; 3Research Group GREIACC, Health Research Institute La Fe, 46026 Valencia, Spain

**Keywords:** leisure activities, leisure time, hospital, humanisation of care, nurse, hospitalised people, patients

## Abstract

Nurses carry out holistic assessments of patients during hospital admission. This assessment includes the need for leisure and recreation. Different intervention programmes have been developed to meet this need. The aim of this study was to investigate hospital leisure intervention programmes described in the literature in order to determine their effects on patient health and highlight the strengths and weaknesses of the programmes as reported by health professionals. A systematic review of articles in English or Spanish published between 2016 and 2022 was carried out. A search was performed in the following databases: CINAHL COMPLETE, PubMed, Cochrane Library and Dialnet and the Virtual Health Library and Web of Science resources. A total of 327 articles were obtained, of which 18 were included in the review. The methodological quality of the articles was assessed using the PRISMA, CASPe and STROBE scales. A total of six hospital-based leisure programmes were identified, including a total of 14 leisure interventions. The activities developed in most of the interventions effectively reduced the levels of anxiety, stress, fear and pain in patients. They also improved factors such as mood, humour, communication, wellbeing, satisfaction and hospital adaptation. Among the main barriers to implementing hospital leisure activities is the need for more training, time and adequate spaces for them develop. Health professionals consider it beneficial for the patient to develop leisure interventions in the hospital.

## 1. Introduction

Hospitalisation is a stressful and vulnerable experience in terms of physical and mental health [1,2]. Adult hospitalised patients are at high risk of deterioration of health due to inactivity during their hospitalisation [1]. Similarly, the hospitalisation process in children leads to an imbalance in their development due to a series of stressful and painful situations involving the interruption of their schooling and social life [2]. In order to promote hospital adaptation, a holistic, interdisciplinary and humanised approach to care needs to be incorporated [3], where spaces for leisure, activities, entertainment and socialising are provided [4].

The participation of hospitalised patients in leisure activities is considered part of comprehensive patient care to maintain their physical and mental wellbeing [1]. Therefore, to improve patient-centred care and contribute to a better hospital experience, nurses must support the promotion of leisure and the management of hospital resources [3]. According to Virginia Henderson’s theory, leisure and recreation are basic needs that must be met during care practice [5]. Nowadays, in most cases, different patient associations and/or non-governmental organisations are in charge of covering this psychosocial care focused on leisure and recreation in hospitals [6]. This is why the WHO recommends paying more attention to this need, especially in the European region. To this end, it has issued the “Declaration on Promoting Patients’ Rights in Europe” [7], which serves as a model for other WHO regions [8]. Likewise, the “American Nurses Association” [9] includes in its Code of Ethics the commitment to meet all needs, and the “American Hospital Association” [10] shows the duty of healthcare professionals to achieve high-quality care.

Currently, few studies analyse patients’ leisure experiences in hospitals [11]. In the United States, for example, Fuqua, in 2012, highlighted the benefits of prescribing television viewing as a recreational activity during hospitalisation, with these including relaxation and reduced stress, anxiety and boredom [4]. However, in the UK, the study by Papaspyros et al., in 2008, showed that certain in-hospital leisure activities that can be carried out from the bedside, such as playing games, surfing the web, receiving phone calls, sending emails, watching TV and listening to the radio could hurt post-cardiac surgery patient ambulation and could contribute to increased hospital stays [12]. Similarly, in the UK, patients’ perceptions of hospital environments were explored in two consecutive studies by Douglas and Douglas, in 2004 and 2005 [13,14]. In these studies, patients identified the need for personal space and resources for recreation and leisure, indicating that the design of the hospital environment can have a significant therapeutic effect on the satisfaction and wellbeing of patients and their families. On the other hand, in Finland, Pelander and Leino-Kilpi, in 2010, conducted a study on children, suggesting that leisure could reduce pain, fear and worry [15].

Therefore, it was necessary to conduct a literature review with the aim of: identifying hospital leisure programmes, determining which activities in the programmes have proven to be effective in improving the health and wellbeing of patients, and establishing the strengths and weaknesses of hospital leisure programmes as pointed out by healthcare professionals.

## 2. Material and Methods

### 2.1. Design

A qualitative systematic review was conducted between February 2021 and October 2022, following the guidelines of the Preferred Reporting Items for Systematic Reviews and Meta-Analyses statement (PRISMA) [16]. The review was conducted using Cochrane guidelines, taking into consideration the Joanna Briggs Institute programme [17,18].

The research question was described by the acronym PIO (patient, intervention and outcome). In this systematic review, the following research question was established: What is the effect of hospital leisure programmes carried out during the leisure time of hospitalised patients?

### 2.2. Search Strategy

First of all, the Medical Subject Headings (MeSH) [19] and the Descriptors of Health Sciences (DeCS) [20] thesauri were identified: “leisure activities”, “actividades recreativas”, “play therapy”, “ludoterapia”, “bibliotherapy”, “biblioterapia”, “art therapy”, “terapia con arte”, “musicteraphy”, “musicoterapia”, “hospitals” and “hospitales”. In addition, the following keywords were also identified from synonyms and MeSH and DeCS-related terms: “lively”, “animación”, “entertainment”, “entretenimiento”, “laughter therapy”, “risoterapia”, “clown” and “payaso”.

Secondly, two researchers independently searched the following electronic databases: CINAHL COMPLETE, PubMed, Cochrane Library, Dialnet, Virtual Health Library and Web of Science. Table 1 shows the search strategies used in each database by linking the descriptors and/or keywords using the Boolean operators “OR” and “AND”. Table A1 shows the most accurate search strategy performed in each database.

### 2.3. Selection Criteria

We included articles meeting our objectives published between 2016 and October 2022 written in English and Spanish.

We excluded articles that addressed the effect of hospital leisure only on a specific disease or in a narrow age range or that addressed the impact of hospital leisure during interventions and not during leisure time.

### 2.4. The Study Selection Process

First, the records obtained from each database were dumped into a bibliographic manager (*n* = 327), whereby 35 duplicate articles were identified and removed.

Secondly, two independent researchers screened the records obtained based on the title and abstract (*n* = 292) to verify compliance with the selection criteria.

A total of 167 articles were selected and retrieved in full text, of which three articles could not be recovered.

Finally, 164 articles were thoroughly reviewed in their entirety to determine their inclusion in the review. A total of 20 papers were selected for a review of their methodological quality, of which 17 met the stipulated methodological quality. In addition, one new study was identified through a reverse search from other methods that met the selection criteria. Thus, a total of 18 articles were selected.

### 2.5. Research Variables

The information obtained was classified according to the variables: hospital leisure programmes, the effectiveness of the leisure activities included in the programmes and the strengths and weaknesses of hospital leisure programmes as reported by health professionals.

### 2.6. Methodological Quality, Level of Evidence and Grade of Recommendation

The methodological quality was established based on the critical reading and assessment of the selected articles. For this purpose, two independent researchers used the tools Preferred Reporting Items for Systematic Reviews and Meta-Analyses (PRISMA) [16], Critical Appraisal Skills Programme Español (CASPe) [21] and Strengthening the Reporting of Observational Studies in Epidemiology (STROBE) [22]. A minimum score of 25 points was set for PRISMA, 7 points for CASPe and 18 points for STROBE.

The level of evidence and grade of recommendation was established on the scale of the Scottish Intercollegiate Guidelines Network (SIGN) [23].

## 3. Results

A total of 18 articles were included in the review. The flow chart of the study selection process is shown in Figure 1. Regarding the design of the articles, 17% (*n* = 3) are systematic reviews, 5.5% (*n* = 1) is represented by a cross-design experimental study and another 5.5% (*n* = 1) a quasi-experimental mixed methods study, 50% (*n* = 9) are observational studies and 22% (*n* = 4) are qualitative studies. Table 2 shows the general characteristics and main results of the included studies.

### 3.1. Analysis of Quality

Regarding methodological quality, systematic reviews were found to be of high quality, according to the PRISMA statement. Most articles that were critically read with the CASPe programme scored 9. According to the STROBE guide, the observational studies obtained a score of 19 and 20.

Regarding the level of evidence, most were found to have a level of evidence of 3 and 4, and a grade of recommendation of D, since the studies had a number of limitations regarding the sample, observer bias and the use of non-validated questionnaires. Thus, certain authors indicate uncertainty as to whether their results could be generalisable. Table 2 shows the quality analysis of the selected studies. 

### 3.2. Hospital Leisure Programmes

The results of the selected studies showed that the central hospital leisure interventions are grouped into: music, art, humour, electronic, accompanying and other types of programmes (Table 3).

The music programmes included passive or active music interventions [24,25,26]. The passive music interventions consisted of assisted relaxation activities and the synchronisation of breathing [24]. The active music interventions, on the other hand, consisted of singing activities, songwriting, playing instruments [24] and using live music in conjunction with guided images [25].

In the study by Ford et al. [26], music and art interventions were implemented, including music listening and personalised songwriting activities with musicians and groups and art activities for patients and family members such as painting, plasticine modelling, printing, collage and drawing. Similarly, in another study, activities such as making jewellery, sun catchers and chimes were carried out as art therapy [27].

In humour programmes, it was noted that laughter therapy includes a wide range of activities such as singing funny songs, laughing for diversion, stretching, playing with hands and dance routines, laughing exercises, healthy clapping and laughing aloud [28]. Several studies [29,30] reported that the activities and skills performed by the hospital clown were interdisciplinary, encompassing humour, drama, music, dance and play.

Regarding electronic media-based programmes, leisure activities associated with electronic devices such as tablets [31] and computers [32] with internet access were found.

The accompaniment programmes are divided into two types of interventions: animal or personal accompaniment, where patients themselves, professionals or other external groups may be involved [32,33].

Other innovative activities mentioned in the studies include therapeutic play, acupuncture and animal entertainment, among others [32,33].

### 3.3. Effectiveness of the Leisure Activities Included in the Programmes

The leisure activities included in the programmes demonstrate significant changes at the biopsychosocial care level. The effects of the leisure activities are presented in Table 3.

Three articles [24,25,26] observed changes in patients following the implementation of music-related activities. People went from feeling anxious, agitated, restless and in pain to feeling calm, content and comfortable during and after the session [24]. In addition, these musical activities produced relaxation and decreased anxiety [25]. The activities of “listening to live music followed by guided relaxation” and “playing instruments” improved mood and reduced pain in patients [34].

The music therapy sessions also allowed patients to share personal feelings. Moreover, significant results were found in terms of decreased pain, anxiety, respiratory rates and heart rates in both active and passive activities [24]. In addition, music and drama workshops were beneficial for the avoidance of problems in people with mental illness [35]. The participatory art interventions allowed the emotions of joy, laughter, sadness and longing to flow among the patients and provided a sense of pleasure, calm and belonging to the group. It also gave the relatives a sense of comfort that their relative was not alone [26]. Art activities improved mood, anxiety and pain [27]. The leisure interventions had positive effects on patients and brought other benefits, such as refurbishing the rooms and providing access to music or painting [24]. 

Humour was found to be a complementary and alternative medicine (CAM) therapy beneficial for relaxation, increased wellbeing and hospital satisfaction [36] as well as for a reduction in anxiety, stress and fear [29], even in children [30]. In adults in long-stay hospitalisation, laughter therapy leisure activities significantly decreased depression and improved sleep quality [28]. However, the medical clowns had a negative effect on the majority of adults [37], and even a fear of the clown figure was experienced among children [30].

Accompaniment by animals decreased pain and anxiety and increased quality of life, socialisation, hospital adaptation, mobility, collaboration, relaxation, pleasure and fun [32].

Young people and adults affirmed the positive consequences of hospital activity interventions, such as: improving mood, reducing stress, promoting communication, relieving the companion and adapting the patient to the hospital [33].

Access to a tablet during patients’ hospitalisation increased patient engagement and perceived satisfaction with their care and hospitalisation experience [31].

### 3.4. Strengths and Weaknesses Reported by Health Professionals about Hospital Leisure Programmes

Healthcare professionals perceived that the leisure activities carried out in hospitals during patients’ free time have strengths and weaknesses.

Most of the nurses who responded to the questionnaire in the study by Baltacı Göktaş et al. in 2017 [38] mentioned that they were aware of the positive health-related effects of music and its role in regulating vital signs. However, a large proportion affirmed that they do not have adequate training, time or working conditions to carry out this resource.

The nurses explained that artistic interventions were necessary to provide more personalised, person-centred care rather than task-based care. Thus, the nurses supported the benefits of these activities and that the activities allowed them to get to know people better, provoking in them a sense of enjoyment and forgetfulness [26].

Regarding the leisure interventions provided by clowns, most health professionals perceived these activities as beneficial and helpful in reducing anxiety levels [30,37,39,40], taking into account that they did not need to interfere with the medical team [40,41]. Although the clowns were reported to be mostly only placed in paediatric wards [37], nurses felt reassured that the child was being accompanied by the clown [30] and noted improved adaptation to the hospital environment and family collaboration in treatment [39]. It was noted that the main barrier to clown availability was financial. The presence of clowns was found to be cost-effective compared to anaesthetics [40]. Paediatricians reported positive experiences with hospital clown interactions; however, they expressed doubts about the benefits of hospital clowning [41].

As a means of coping with pain and alleviating suffering, Bermúdez Rey et al., in 2019, highlighted in their study the opinion of healthcare staff, who concluded that it was important to carry out leisure programmes to alleviate suffering, cope with pain and humanise hospitals [33]. Table 3 shows the general characteristics of the different studies selected.

## 4. Discussion

Studies show that most adult hospitalised patients spend their free time during hospitalisation watching television, reading or listening to music, etc., although they affirm that patients would prefer other leisure activities if they were offered in the hospital [33]. In the child population, studies show that children routinely carry out similar activities in their bedrooms [42].

According to the articles reviewed in this study, regarding age and the demand for leisure activities, it can be observed that the older the age, the less demand there is for leisure activities. This may be because older people are more disinterested in leisure activities because they believe these activities do not bring about changes in them. This negative view may be related to their perception of health, increased pathologies, the tendency towards chronicity, the risk of disability and the vulnerability of the elderly. Bermúdez Rey et al., in 2019, argued that implementing leisure activities could reverse this negative view [33]. These results coincide with the study carried out by the same author at the Hospital La Fe in Valencia [43]. However, another study conducted with older people found that people and their relatives reported that they were satisfied with participatory art, music and comedy activities [27], with their mood improving after the activity [26].

Studies by Boulayoune et al., published in 2020 [30], and Sridharan and Sivaramakrishnan, published in 2016 [29], indicate the benefits of hospital entertainment for children, with indirect benefits for their parents. These findings are consistent with an earlier study, where parents rated the effect of hospital entertainment very positively, stating that it gave them peace of mind [42]. In another study exploring parents’ views on volunteering in hospital activities, 100% of parents supported this intervention [6]. In addition, it has been shown that the inclusion of relatives in different recreational activities provides a higher level of satisfaction [24,27].

Regarding the effects of hospital leisure activities, it has been observed that, as a result of leisure activities the hospital environment becomes a less clinical and more welcoming place for patients and relatives. Participants in the 2018 study by Ford et al. [26] affirmed a sense of pleasure, calm and belonging when decorating the unit. Similarly, in a previous study [42], both children and parents admitted that they would change the decoration of their room, with more than half of the sample stating that they would prefer individual rooms with more colour, games, pictures and space for education, leisure and play as well as access to outdoor spaces.

Another effect of leisure use worth mentioning is the way its increase patients’ abilities to cope with pain [24,25,26,29,34]. In addition, the use of leisure activities showed that by alleviating pain, the development of other problems such as depression, insomnia, weight gain, reduced mobility and reduced socialisation was avoided. This is explained by that fact that when investing time in a satisfying leisure activity, the person’s attention is diverted away from the focus of the pain [43].

The leisure programmes offered in this research include music [24,25,26,34], art [26,27], humour [28,29,30], electronic entertainment [31,32], companionship [32,33] and other programmes [32,33]. In addition, one study is developing a virtual reality entertainment programme which can connect people from adjacent rooms to reduce stress and boredom which could involve all hospitalised patients, including those isolated by infection or immunosuppression. It would also reduce hospital costs, since it would not be necessary to hire staff to carry it out [44].

In the studies reviewed, it was observed that the activities targeted at people of all ages are: active music sessions, painting, chatting, walking, reading, listening to music, hobbies, playing therapeutic games, massages and active music sessions. Passive music sessions have only been studied in patients over 10 years of age. Hospital clowning is aimed at children and their carers, while laughter therapy sessions are aimed at under-18s, along with the use of computers and animal companionship. The use of tablets is aimed at over 18s. Activities for adults include: bead-making and handicrafts, professional accompaniment, guided discussion groups, board games, parties, visits by adult patients to paediatric patients and workshops of different types. Drawing and room decoration activities are aimed at the elderly.

Concerning musical activities, the studies by Scheufler et al. [25] and West and Silverman in 2020 [34] supported the same three interventions: listening to the hospitalised patients’ favourite music, the active creation of music with instruments and listening to live music together with guided imagery. From this research, it could be seen that music takes a leading role in inhibiting problems, enhancing relaxation and reducing anxiety. Most participants listened to their favourite live music (passive music) [24,25,34]. This activity resulted in more relaxation and decreased somatic anxiety [25]. In addition, music and drama workshops were beneficial for the avoidance of problems in people with mental illness [35].

Clown activities are closely related to laughter therapy, which reduces anxiety, stress and fear in children and parents [30]. Hospital clowns have interdisciplinary skills such as magic, theatre, music, play, dance and humour, which are therapeutically valuable activities. Play is the therapeutic tool that provides the most significant sense of security for the child [29,30]. However, depending on the patient’s age and affective state, satisfaction with the clown’s intervention varies [37,39]. It is therefore necessary to assess and adapt the interventions for each hospitalised patient according to their needs and the state of their illness [30].

However, certain people would be willing to pay for humour therapy to achieve stress management [36]. Losada et al., in 2019 [45], affirmed that now is the right moment to make a paradigm shift, using laughter to help tolerate problems. However, he warned that this therapy should never be forced [45], as cases have been found where the child does not enjoy the clown’s visit [46].

Likewise, healthcare staff regard the hospital clown very positively; they admit that the clown is useful and relieves anxiety in children and parents [30,39,40] and believe that this resource should not only be in paediatric wards [37]. However, in a study where the opinions of paediatricians were collected, they affirmed that they needed to learn the exact benefits of this resource and set conditions to collaborate with them in the hospital [41]. It follows that for the hospital clown to provide relief and entertainment to the hospitalised patients [30,39,41], it is essential that there is good communication with the healthcare team and that the clown does not interfere with their work [40,41].

The perception of healthcare staff regarding the implementation of play activities in the hospital is mostly positive, with them admitting that they are necessary activities to achieve more personalised and humane care [27,40]. Similarly, it is essential to note that nurses are interested in learning techniques and skills to entertain and improve patient satisfaction [27]. In addition, they admit that through these activities they get to know their patients better and feel more comforted by them [27,39]. However, according to certain nurses, implementing play activities in the hospital has limitations such as a lack of resources, the fear of professionals and a lack of time and training [38]. They also admit that for a successful art programme, it is necessary to personalise the activities according to the abilities and needs of the patients [26].

### Limitations of the Review

The limitations of this review include the need for more studies with information on the subject, the impossibility of containing all the records identified due to difficulties in accessing a number of manuscripts and the small sample size in most of the studies, which had an impact on external validity. Therefore, multi-centre research with robust designs and an adequate sample size is needed to explore the efficacy of hospital leisure activities. On the other hand, it would be interesting to analyse who is the most suitable person to carry out this type of activity, whether the health professionals themselves or, on the contrary, people from outside the institution. Likewise, it would be helpful to analyse how implementing these activities is perceived in the context of the dynamics of the healthcare team.

## 5. Conclusions

Leisure interventions carried out during leisure time positively impact and promote the wellbeing of hospitalised patients. Leisure proposals during the hospital stay revolve around music, interactions with the hospital clown, art activities, the use of electronic tablets and various workshops.

Hospital leisure effectively reduces pain, stress, fear and anxiety, improves mood, humour and communication and promotes adaptation to the hospital, providing a greater sense of calm, wellbeing, belonging and satisfaction. Leisure programmes have the same effect on children, adults and the elderly, with the exception of the hospital clown, which has a greater effect on children.

Healthcare staff are willing to learn and carry out recreational activities and acknowledge that they help to achieve more personalised and humane care.

## Figures and Tables

**Figure 1 ijerph-20-03268-f001:**
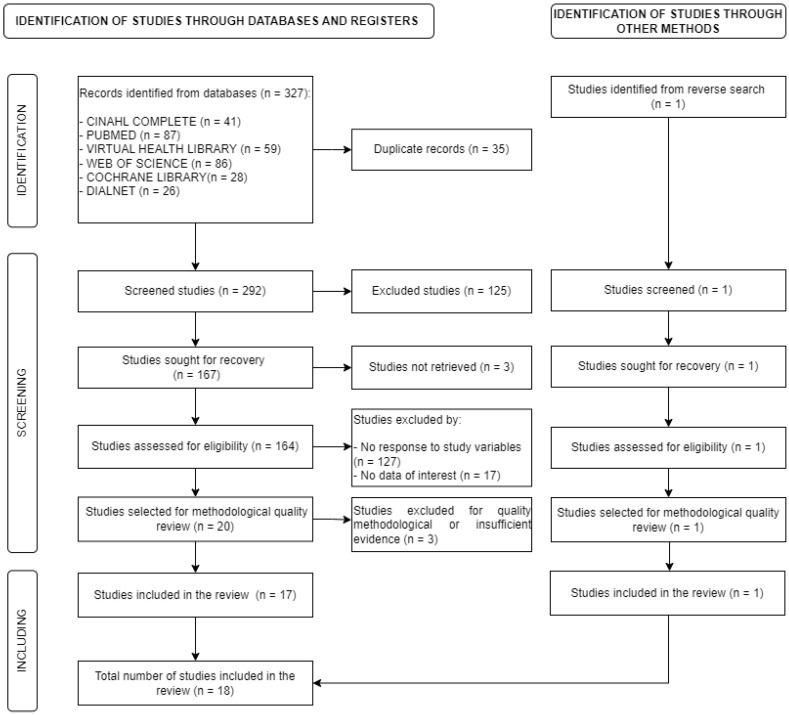
Flow chart of the study selection process.

**Table 1 ijerph-20-03268-t001:** Database search strategy.

Database	Search String	Records Obtained
CINAHL	(leisure activities OR animation OR play therapy OR clown OR bibliotherapy OR art therapy OR music therapy) AND (hospitals) (lively OR entertainment OR laughter therapy) AND (hospitals)	41
PUBMED		87
VIRTUAL HEALTH LIBRARY		59
WEB OF SCIENCE		86
COCHRANE LIBRARY		28
DIALNET		26

**Table 2 ijerph-20-03268-t002:** General characteristics and main results of the included studies.

Author Year Country	Types of Study Sample	Objective Intervention	Study Variable	Results and Conclusions	Quality, LE * and GR *
Fallek et al. [24] 2020 United States	Quasi-experimental pre-post mixed methods study. A total of 150 patients over 18 years of age at Montefiore Medical Center from four inpatient units: palliative care, transplant, medical intensive care and general medicine.	To assess the effects of music therapy. To assess the feasibility of providing music therapy in the units and the patients’ interest, receptivity and satisfaction. Active participation, passive participation or a combination of both.	Anxiety and pain. Patient interest, responsiveness and satisfaction.	The preferred musical activities were passive rather than active and produced decreased pain, anxiety, respiratory and heart rates. Shorter sessions reduced anxiety more in younger people. However, longer sessions reduced anxiety more in the middle-aged group. All in all, the activity was accessible, feasible and produced wellbeing. In addition, patients and their families were interested in the passive activities.	CASPe 9/10 SIGN LE: 3 GR: D
Scheufler et al. [25] 2020 United States	Experimental study of a three-treatment, three-intervention crossover design. A total of 40 females and 8 males aged 10–18 years with amplified pain who started an IIPT programme in a children’s hospital in the Midwestern United States.	To compare three music therapy interventions for anxiety and relaxation in young people with amplified pain. State Trait Inventory for Cognitive and Somatic Anxiety for Children and EVA scale.	Music therapy interventions. Levels of anxiety and amplified pain.	Live music was the intervention that produced the most relaxation and reduced somatic anxiety the most, along with patient-selected music. Cognitive anxiety was similarly improved in all three interventions. Consequently, all three interventions resulted in more relaxation, less somatic anxiety and less cognitive anxiety.	CASPe 7/10 SIGN LE: 1- GR: B
Ford et al. [26] 2018 Australia	Qualitative study. A total of 22 patients with dementia or delirium, or both, in an intensive care unit for the elderly in a general hospital. In all, 18 healthcare staff: medical staff (*n* = 3); nurses (*n* = 7); allied health (*n* = 4); a medical student (*n* = 1) and ward support staff (*n* = 3).	To evaluate the impact of art in health programmes delivered within an acute older person’s unit. Observation of arts activities, semi-structured interviews with patients and relatives and focus groups with staff.	Emotions. Staff perspective. Influence of the programme on the environment.	There were participatory art activities, personalised music, songs and artwork and the liveliness of the unit’s environment. The artistic interventions brought out emotions, the relatives felt comforted by the activities and the staff got to know the patients better. In addition, the different leisure activities were positive received by the participants, and the team found these activities useful for the inpatients.	CASPe 9/10 SIGN: LE: 4 GR: D
Shella [27] 2018 United States	Longitudinal observational descriptive study. A total of 166 females and 29 males, aged 28–62 years, referred from a large urban teaching hospital with diagnoses of: cancer, neurological diseases, gastrointestinal problems, cardiac and vascular diseases, transplantation, post-surgical and orthopaedic procedures.	Perform art therapy to assess: mood, anxiety and pain. Roger’s Sad and Happy Face Scale.	Types of art activities. Levels of pain, mood and anxiety between the beginning and the end of art therapy sessions.	The types of art activities included mainly painting and pearls. In conclusion, after participating in an art therapy session, mood was improved and anxiety and pain were reduced.	STROBE 19/22 SIGN LE: 3 GR: D
Han et al. [28] 2017 Korea	Observational analytical studies: a case-control study. A total of 42 residents from two long-term care hospitals.	To investigate the effects of laughter therapy on depression and sleep among patients in two long-term care hospitals. MMSE-K (cognition), K-ADL (quality of life), GDSSF-K (depression) and PSQI-K (sleep) scales.	Effects of laughter therapy on depression and sleep quality in adults living in a long-stay hospital.	The laughter therapy protocol consisted of singing funny songs, laughing for diversion, stretching, playing with hands and dance routines, laughing exercises, healthy clapping and laughing aloud. The findings showed that depression and sleep improved in the treatment group compared to the comparison group. To improve depression and sleep among long-term care hospitals, offering laughter therapy to strengthen physical activities might benefit patients.	STROBE 21/22 SIGN LE: 2+ GR: C
Sridharan et al. [29] 2016 Fiyi	Systematic review and meta-analysis. A total of 18 randomised controlled trials (review) and 15 for meta-analysis. Children who were admitted to hospitalisation or underwent invasive procedures and their caregivers.	To compare the clinical utility of hospital clowns compared to the standard care in relieving fear, anxiety and pain in children. Review of studies.	Anxiety, fear and pain in children with or without the clown. Anxiety of parents.	It was observed that medical clowns relieved pain, fear, anxiety and stress and increased morale. They also provided support for parents. Clowns performing activities in hospitals are therapeutically helpful to hospitalised children and their parents.	PRISMA 25/27 SIGN LE: 2++ GR: B
Boulayoune et al. [30] 2020 Spain	Systematic review. A total of 15 articles on paediatric population.	To review the physiological and psychological effects of laughter therapy in the paediatric population. Review of studies.	Anxiety, stress and fear. Perception of the hospital setting. Precursor activities of laughter therapy.	The humour reduced anxiety, stress and fear in hospitalised children and their parents. Health care staff affirmed that clowns are helpful. The clowns used different methods to entertain the child. Therefore, the humour was beneficial and helped the hospitalised child’s adjustment.	PRISMA 27/27 SIGN LE: 2++ GR: B
Vink et al. [31] 2019 United States	Observational study An analytical cross-sectional case-control study. A total of 117 074 patients admitted to the six hospitals of the medical centre. Patients were excluded who were under 18 years old, who could not speak or read English or who were involuntarily detained, as were patients with less than one night’s stay in the hospital, patients with a principal psychiatric diagnosis or patients who were not alive at the time of discharge.	To explore the relationship between hospital access to tablets and patient satisfaction with hospital care. HCAHPS survey.	Patients’ perspectives on their healthcare experience during their hospital stay.	Patients who received tablets scored higher on the HCAHPS. Thus, the provision of tablets to hospital inpatients improved aspects of their care, experience and patient engagement, with this associated with greater satisfaction with their hospital stay.	STROBE 19/22 SIGN LE: 2+ GR: C
Barros et al. [32] 2021 Portugal	Systematic review. A total of 14 articles. Patients under 18 and their family	To identify the process of child and family adaptation to hospitalisation and to map nursing interventions that promote child/young person/family adaptation to hospitalisation. Review of studies.	The child’s adaptation to hospitalisation. Parents’ adaptation to hospitalisation. Strategies that promote adaptation to hospitalisation.	Interventions for children focus on strengthening coping mechanisms and increasing security. Interventions such as therapeutic play, anticipatory information, relaxation techniques, distraction, humour, music therapy, adaptation kits, therapeutic groups and hope-promoting strategies were emphasised. Nursing interventions decrease the anxiety and stress of the child/family, increasing their ability to receive information and participate in care and decisions.	PRISMA 25/27 SIGN LE: 2++ GR: B
Bermúdez Rey et al. [33] 2019 Spain	Observational, descriptive, cross-sectional study. A total of 46 patients aged between 31 and 70, 35 accompanying persons and 26 healthcare staff, with patients in good physical and mental condition at the Marqués de Valdecilla University Hospital in Santander (Cantabria).	To study the perception of the patients, families and professionals about the need to occupy the free time of hospitalised patients. Questionnaire of closed questions.	Occupation of patients’ leisure time in hospitals.	Patients occupied their time in their room but would have liked to participate in the proposed leisure activities. The health care staff supported leisure time. In conclusion, there were significant differences in the need for leisure time according to age and length of stay in the hospital. Young and adult patients and healthcare staff supported the leisure programmes.	STROBE 20/22 SIGN LE: 3 GR: D
West et al. [34] 2020 United States	Observational study, descriptive longitudinal study. The 29 participants were adults receiving care on a general neuroscience unit within a University-affiliated hospital in the Midwestern region of the United States.	To study the frequency of the user’s choice of the music therapy technique. Additionally, to study the effects on mood and pain in adults receiving music therapy. Mood (QMS) and pain (Likert) scales.	Music therapy preferences. Effectiveness of music therapy.	Most participants listened to their favourite live music, followed by guided relaxation and the playing of instruments. No significant differences appeared between the interventions; all had positive changes in mood and pain. Thus, hospitalised adults preferred to engage with music passively rather than actively.	STROBE 21/22 SIGN LE: 3 GR: D
Ørjasæter et al. [35] 2018 Norway	Qualitative study, hermeneutic-phenomenological analysis. A total of 11 adult participants in a music and theater workshop carried out in a Norwegian mental health hospital.	To explore experiences of participation in music and drama among people with long-term mental illness. Open interviews.	Participation in music and drama programmes. Perception of music and drama programmes.	The music and theatre workshops made people with mental illnesses not think about their problems. The patients reported that it serves as mental encouragement to regain meaningful activities in daily life. They also felt a sense of achievement and hope for a better future. It was concluded that music and drama workshops for hospitalised patients positively affected the participants’ motivation to live and discover life outside the institution.	CASPe 10/10 SIGN LE: 4 GR: D
Montross-Thomas et al. [36] 2017 United States	Cross-sectional descriptive observational study. Adult patients (*n* = 100), ranging in age from 19–95 years, were recruited during their hospitalization in the University of California, San Diego, Healthcare System.	To show perspectives on complementary and alternative medicine (CAM) services. Closed response interview.	CAM preferences. Declared willingness to pay for complementary therapies.	Useful complementary therapies they would be willing to pay for were: healthy food, massage therapy and humour therapy. The inpatients felt that CAM treatments would provide relaxation and increase their wellbeing and hospital satisfaction. In short, patients are interested in CAM services during their hospitalisation.	STROBE 19/22 SIGN LE: 3 GR: D
Efrat-Triester et al. [37] 2021 Israel	Cross-sectional descriptive observational study. A total of 495 participants, made up of 88 healthcare workers, 20 medical clowns and 387 health consumers.	Exploring the usefulness of clowns. Survey.	Perceptions of healthcare workers. Perceptions of medical clowns. Perceptions of patients.	Healthcare staff reported that medical clowns increase children’s satisfaction more than adults. Medical clowns felt that they should be more in demand. The patients most satisfied by the clown’s performance were those under 21.6 years of age. Ultimately, the usefulness of medical clowns in improving patient satisfaction and reducing aggression depended on age.	STROBE 20/22 SIGN LE: 3 GR: D
Baltacı GÖKTAŞ et al. [38] 2017 Turkey	Cross-sectional descriptive observational study. A total of 225 nurses and midwives	To determine the knowledge, understanding, behaviour and practices of music therapy among nurses and midwives. Survey.	Knowledge and opinions about music therapy. Effect of music therapy on vital signs. Use of music therapy.	Most nurses were aware of the benefits of music for hospitalised patients, although they had not been educated about it. Only 7% used this therapeutic tool and more than half reported not having the time to apply it. In short, despite being aware of the benefits of music for patients, the nurses affirmed that they did not feel trained in applying this resource.	STROBE 18/22 SIGN LE: 3 GR: D
Masetti et al. [39] 2019 Brazil	Cross-sectional descriptive observational study. A total of 567 professionals from 13 public hospitals	To analyse health professionals’ perception of “Doutores da Alegria’s” interventions using a questionnaire. Questionnaire.	Permanence of interventions. Intrapersonal and interpersonal effects on health staff, children and family members.	The “Doutores da Alegria” positively impacted the child’s adaptation. The health professionals affirmed that they feel more satisfied when learning how to approach children and their families. Therefore, the perception of the “Doutores da Alegria” and their interventions were positive.	STROBE 20/22 SIGN LE: 3 GR: D
Gomberg et al. [40] 2020 Israel	Qualitative grounded theory study. A total of 35 subjects were interviewed with 35 doctors, nurses and technicians at the Davidoff oncology ward of Beilinson Hospital in Petah Tikva, Israel.	To explore hospital staff perspectives on the impact of medical clowning. Individual semi-structured interviews.	Perspectives of doctors, nurses and technicians on the medical clown.	Healthcare staff reported cost-saving measures for the hospital by having clowns. Other benefits described were increases in staff efficiency, improved patient outcomes and reduced stress on the medical staff. In conclusion, many unpublished benefits of medical clowns were described.	CASPe 9/10 SIGN LE: 4 GR: D
Van Venrooijy et al. [41] 2017 Netherlands	Qualitative study. A total of 14 paediatricians of the paediatric departments of Haga Hospital The Hague and Leiden University Medical Center, the Netherlands, to evaluate the effect of hospital clowning in children between 3 and 12 years of age.	To investigate the current situation of hospital clowns from the perspective of paediatricians and paediatric residents. Focus groups.	Perception of paediatricians.	Most paediatricians reported favourable experiences with the hospital clown, although many were unaware of the potential benefits. They also described various conditions for collaborating with them. The study concluded that there were controversial opinions about clowns in hospitals.	CASPe 8/10 SIGN LE: 4 GR: D

* LE: Level of evidence, GR: grade of recommendation.

**Table 3 ijerph-20-03268-t003:** Summary of the relationship between leisure programmes, interventions and activities and their effects.

Programme	Interventions	Age	Activities	Effects
Music	Passive music sessions [24,25]	Over 10	Music-assisted relaxation. Synchronisation of breathing with the music of different genres (jazz, salsa, pop, etc.) or music from the favourite countries of the people assisted.	Personal feelings are shared. Avoidance of the problems of people with mental illnesses. Increased peace of mind, joy and comfort. Improvement in mood. Decreased pain and anxiety, breathing rate and heart rate.
	Active music sessions [24,25,26,32]	For all ages	Singing. Songwriting. Playing instruments with family members. Listening to and the singing of individualised music: organisation of 25 songs of different musical genres according to the person’s preferences and the singing of meaningful lyrics. Active creation of music and active improvisation of instruments: the patient could show emotions thanks to the xylophone’s rhythm. Creation of personalised music and songs with musicians or in a group. Using live music together with guided images: the patient chose a comfortable environment and the music therapist personalised his images. The music therapist played live music on the guitar while the patient visualised the images.	
Art	Painting [26,27,32]	For all ages	Acrylic on objects, canvas or paper, alcohol-based inks on tile and watercolour on paper.	Feeling of pleasure, calm and belonging to the group. Comforted family members. Feeling of enjoyment and forgetfulness. Improved mood. Flow of emotions. Decreased anxiety and pain.
	Beads [27]	Those 28–62 years old	Making jewellery, sun catchers and bells from wires or ropes.	
	Handicraft works [27]		Making papier-mâché, making/decorating wooden models, collages, plasticine modelling and drawing.	
	Room decoration [26]	Elderly people	The room was refurbished and painted, with access to music. Artwork by both patients and staff was displayed. Patients’ artwork was placed in public areas and on their beds, personalising their space.	
	Illustration/drawing for the patient [26]		The artist made a drawing depicting the memories that the patient had transmitted to him.	
Humour	Hospital clown [29,30]	Children and their caregivers	Humour. Theatre. Music. Dance. Children’s stories. Magic tricks.	Relaxation. Increased wellbeing. Hospital satisfaction. Reduction in anxiety, stress and fear.
	Laughter therapy sessions [28,32]	Under 18	The laughter therapy programme consisted of two 40-min sessions per week. For the first 5 min, patients were motivated to participate and to increase the feeling of closeness through hugs, compliments and handshakes. The following 25 min were spent on the following activities: singing funny songs, complimenting the positive qualities of other participants, laughing, stretching, playing hand games and dancing. The session ended with 10 min of relaxation, maintaining a positive mood, meditation and expressing thoughts and feelings.	Decrease in depression. Increased quality of sleep.
Electronic	Tablets [31]	Over 18	Have access to a tablet during hospitalisation for consultations, personal health information or entertainment.	Increased patient participation. Perceived satisfaction with their care and hospital experience.
	Computers [32]	Under 18	Video games. Computer games.	
Accompaniment	Accompaniment by animals [32]	Those under 18	Accompaniment by animals.	Reduction in pain and anxiety. Increased quality of life. Socialisation. Hospital adaptation. Mobility. Collaboration. Relaxation. Pleasure. Fun
	Accompaniment by people [33]	Adults	Accompaniment by professionals. Guided discussion groups. Board games. Celebration of parties. Visits from adult patients to paediatric patients. Workshops of different types.	Development of adaptation strategies.
Others	[28,33]	For all ages	Chatting. Strolling. Reading. Listening to music. Doing pastimes. Playing therapeutic games. Massage (acupuncture). Participating in hospital magazines.	

## Data Availability

Not applicable.

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
