# Peer review of "Leisure Programmes in Hospitalised People: A Systematic Review"

_ijerph, 2023, doi:10.3390/ijerph20043268_

Round 1

Reviewer 1 Report

Dear authors,

I think your manuscript is very well thought out and very well presented.

I really liked the topic and the importance you gave it.

I suggest only a few small things that I consider appropriate for correction.

-          - The last three sentences in the abstract seem to repeat the previous part of the text.

-          - A space is missing between words and references throughout the entire manuscript.

-          - Please clarify what "one new study was identified from other methods" means, which other methods?

-         -  Line 122: is “y” a mistake or?

-         -  I believe that section 3.4. Analysis of quality should have been shown a little earlier in the results. Maybe before Hospital leisure programs.

-         -  References should be edited and standardized according to the journal's instructions.

Author Response

Dear reviewer,

We would like to thank you for your contributions, which undoubtedly contribute to improving the quality of the manuscript. We have taken into account all your suggestions which have been included in the document.

  1. The last three sentences in the abstract seem to repeat the previous part of the text.

Thank you very much for your suggestion. “There is a wide variety of hospital leisure programmes and interventions. Most interventions have positive effects on the health and well-being of patients” has been deleted. But we have considered it appropriate to keep the last sentence because it states the opinion of health professionals “Health professionals consider it beneficial for the patient to develop leisure interventions in the hospital” (line 24-25).

  1. A space is missing between words and references throughout the entire manuscript.

Thank you very much for your contribution. Accepted and corrected.

  1. Please clarify what "one new study was identified from other methods" means, which other methods?

Thank you, the article was selected by reverse search. This concept is added in the manuscript for clarification (line 111-112).

  1. Line 122: is “y” a mistake or?

This is certainly a mistake. Thank you very much, it has been corrected.

  1. I believe that section 3.4. Analysis of quality should have been shown a little earlier in the results. Maybe before Hospital leisure programs.

Thank you very much, the section has been moved according to your recommendation. (line 139).

  1. References should be edited and standardized according to the journal's instructions.

Thank you very much, the references have been corrected and adjusted to the journal's standards.

Reviewer 2 Report

The prospect of well-organized rest time during hospitalization is very important for every patient. Regardless of age, it improves his health. I congratulate the Authors on their choice of topic and well described research review. However, I would like some additions:

Line 29 Keywords: propose to add the word "patient" as recreational programs are performed on a hospitalized patient

Line 136 Table 2 sub-heading „Types of study Sample” Is anything known about these study patients? Did the study involve children or adults? what wards were they hospitalized in? for what diseases?

Line 180 Table 3 When summarizing the relationship between leisure programmes, interventions and activities and their effects, are you able to add a column for the age of the patients? Were these effects the same in children, adults, or older patients?

Linia 279-288 which of these programs are dedicated to children and which to adults and the elderly?

Line 326 study limitations exclude from discussion and discuss as a separate point "limitations"

Line 340-342, could you say at this point in the summary whether it has the same effect on adult and older children?

Author Response

Dear reviewer,

We would like to thank you for your contributions, which undoubtedly contribute to improving the quality of the manuscript. We have taken into account all your suggestions which have been included in the document.

  1. Line 29 Keywords: propose to add the word "patient" as recreational programs are performed on a hospitalized patient

Thank you very much for the suggestion.The word patient is added to the list of keywords (line 26).

  1. Line 136 Table 2 sub-heading „Types of study Sample” Is anything known about these study patients? Did the study involve children or adults? what wards were they hospitalized in? for what diseases?

Thank you very much. More information on the sample is added (table 2).

  1. Line 180 Table 3 When summarizing the relationship between leisure programmes, interventions and activities and their effects, are you able to add a column for the age of the patients? Were these effects the same in children, adults, or older patients?

Thank you, a column on the age for interventions is added to the table (table 3).

  1. Line 279-288 which of these programs are dedicated to children and which to adults and the elderly?

Thank you very much. We have added a paragraph explaining this question (lines 290-299).

  1. Line 326 study limitations exclude from discussion and discuss as a separate point "limitations"

Thank you. Accepted and corrected (line 340).

  1. Line 340-342, could you say at this point in the summary whether it has the same effect on adult and older children?

Thank you. We have added “Leisure programmes have the same effect on children, adults and the elderly, with the exception of the hospital clown, which has a greater effect on children” (lines 357-359).

Reviewer 3 Report

Dear all:

The article presents a systematic review of the literature on a topic little explored in the literature, it has publication potential, but needs changes.

Introduce the purpose of the study in the abstract and at the end of the introduction.

In line 63 of the introduction, present only the surname of the authors and not the initials of the name.

The research question presented at the end of the introduction should move to the method.

In the method you state that the study was supported by PRISMA - but this is a report guideline and not a methodological framework. Does the systematic review follow the JBI or the Cochrane protocol? What type of systematic review was performed?

Clarify which acronym is used to elaborate the research question and present the inclusion criteria related to the participants, interventions, context. Present the search strategy used, in at least one database, with the association of keywords in natural language and the indexed terms of the respective database.

Figure 1 should be moved to results.

In table 3, associate the interventions with the reference number of the study that refers to them.

Present the limitations associated with the method and methodological procedures.

Author Response

Dear reviewer,

We would like to thank you for your contributions, which undoubtedly contribute to improving the quality of the manuscript. We have taken into account all your suggestions which have been included in the document.

  1. Introduce the purpose of the study in the abstract and at the end of the introduction.

Thank you very much for the suggestion. The objective of the review is added to the summary (line 12). The objective was already described at the end of the introduction (lines 67-71).

  1. In line 63 of the introduction, present only the surname of the authors and not the initials of the name.

Agreed, it has been amended in the text (line 61).

  1. The research question presented at the end of the introduction should move to the method.

Thank you for the input. The research question has been moved on your recommendation. (line 78).

  1. In the method you state that the study was supported by PRISMA - but this is a report guideline and not a methodological framework. Does the systematic review follow the JBI or the Cochrane protocol? What type of systematic review was performed?

The review was conducted using Cochrane guidelines and taking into consideration the Joanna Briggs Institute programme (lines 76-77). The research paper is a qualitative systematic review (line 74).

  1. Clarify which acronym is used to elaborate the research question and present the inclusion criteria related to the participants, interventions, context.

Thank you very much. Accepted and corrected.  The research question was described by the acronym PIO (patient, intervention and outcome) (line 78). The selection criteria are in line 90.

  1. Present the search strategy used, in at least one database, with the association of keywords in natural language and the indexed terms of the respective database.

We have added table 1A (line 94). Table 1A shows the most accurate search strategy performed in each database (line 362).

For example, in CINAHL COMPLETE:

(leisure activities OR animation OR play therapy OR clown OR bibliotherapy OR art therapy OR music therapy) AND (hospitals) in (Title)

Limiters - Date of publication: 20160101-20211231; Language: English, Spanish

Enlargers - Apply Equivalent Subjects

Search Modes - Boolean/Phrase

Results: 40

https://web.p.ebscohost.com/ehost/resultsadvanced?vid=32&sid=e120d7a9-35ff-430e-94f7-59c07cff35bf%40redis&bquery=TI+(leisure+activities+OR+animation+OR+play+therapy+OR+clown+OR+bibliotherapy+OR+art+therapy+OR+music+therapy)+AND+(hospitals)&bdata=JmRiPWNjbSZjbGkwPURUMSZjbHYwPTIwMTYwMS0yMDIyMTImbGFuZz1lcyZ0eXBlPTEmc2VhcmNoTW9kZT1TdGFuZGFyZCZzaXRlPWVob3N0LWxpdmUmc2NvcGU9c2l0ZQ%3d%3d

(animation OR entertainment OR  laughter therapy) AND (hospitals)

Limiters - Date of publication: 20160101-20211231; Language: English, Spanish

Enlargers - Apply Equivalent Subjects

Search Modes - Boolean/Phrase

Full text

Results: 1

https://web.s.ebscohost.com/ehost/resultsadvanced?vid=3&sid=7eaf2845-5130-4372-bbf5-3fb67ebf1b98%40redis&bquery=TI+(animation+OR+entertainment+OR++laughter+therapy)+AND+(hospitals)&bdata=JmRiPWNjbSZjbGkwPUZUJmNsdjA9WSZjbGkxPURUMSZjbHYxPTIwMTYwMS0yMDIyMTAmY2xpMj1MQTk5JmNsdjI9ZW5nJTdlc3BhJmxhbmc9ZXMmdHlwZT0xJnNlYXJjaE1vZGU9U3RhbmRhcmQmc2l0ZT1laG9zdC1saXZlJnNjb3BlPXNpdGU%3d

  1. Figure 1 should be moved to results.

Thank you. Accepted and corrected (line 135).

  1. In table 3, associate the interventions with the reference number of the study that refers to them.

Accepted and corrected (line 193).

  1. Present the limitations associated with the method and methodological procedures.

Thank you. The limitations of the review can be found in line 340.
